# Comparative Phenotyping of Two Commonly Used *Chlamydomonas reinhardtii* Background Strains: CC-1690 (21gr) and CC-5325 (The CLiP Mutant Library Background)

**DOI:** 10.3390/plants11050585

**Published:** 2022-02-22

**Authors:** Ningning Zhang, Leila Pazouki, Huong Nguyen, Sigrid Jacobshagen, Brae M. Bigge, Ming Xia, Erin M. Mattoon, Anastasiya Klebanovych, Maria Sorkin, Dmitri A. Nusinow, Prachee Avasthi, Kirk J. Czymmek, Ru Zhang

**Affiliations:** 1Donald Danforth Plant Science Center, St. Louis, MO 63132, USA; lucy.nnzhang@gmail.com (N.Z.); lpazouki@danforthcenter.org (L.P.); huong.nguyen@ttu.edu (H.N.); cxia@danforthcenter.org (M.X.); emattoon@danforthcenter.org (E.M.M.); aklebanovych@danforthcenter.org (A.K.); mlsorkin@wustl.edu (M.S.); meter@danforthcenter.org (D.A.N.); kczymmek@danforthcenter.org (K.J.C.); 2Department of Biology, Western Kentucky University, Bowling Green, KY 42101, USA; sigrid.jacobshagen@wku.edu; 3Department of Biochemistry and Cell Biology, Geisel School of Medicine at Dartmouth, Hanover, NH 03755, USA; brae.m.bigge.gr@dartmouth.edu (B.M.B.); prachee.avasthi@dartmouth.edu (P.A.); 4Plant and Microbial Biosciences Program, Division of Biology and Biomedical Sciences, Washington University in Saint Louis, St. Louis, MO 63130, USA

**Keywords:** *Chlamydomonas reinhardtii*, CC-1690, CC-5325, Chlamydomonas Library Project (CLiP), photosynthesis, heat tolerance, cell wall, circadian rhythm of phototaxis

## Abstract

The unicellular green alga *Chlamydomonas reinhardtii* is an excellent model organism to investigate many essential cellular processes in photosynthetic eukaryotes. Two commonly used background strains of Chlamydomonas are CC-1690 and CC-5325. CC-1690, also called 21gr, has been used for the Chlamydomonas genome project and several transcriptome analyses. CC-5325 is the background strain for the Chlamydomonas Library Project (CLiP). Photosynthetic performance in CC-5325 has not been evaluated in comparison with CC-1690. Additionally, CC-5325 is often considered to be cell-wall deficient, although detailed analysis is missing. The circadian rhythms in CC-5325 are also unclear. To fill these knowledge gaps and facilitate the use of the CLiP mutant library for various screens, we performed phenotypic comparisons between CC-1690 and CC-5325. Our results showed that CC-5325 grew faster heterotrophically in dark and equally well in mixotrophic liquid medium as compared to CC-1690. CC-5325 had lower photosynthetic efficiency and was more heat-sensitive than CC-1690. Furthermore, CC-5325 had an intact cell wall which had comparable integrity to that in CC-1690 but appeared to have reduced thickness. Additionally, CC-5325 could perform phototaxis, but could not maintain a sustained circadian rhythm of phototaxis as CC1690 did. Finally, in comparison to CC-1690, CC-5325 had longer cilia in the medium with acetate but slower swimming speed in the medium without nitrogen and acetate. Our results will be useful for researchers in the Chlamydomonas community to choose suitable background strains for mutant analysis and employ the CLiP mutant library for genome-wide mutant screens under appropriate conditions, especially in the areas of photosynthesis, thermotolerance, cell wall, and circadian rhythms.

## 1. Introduction

The unicellular green alga *Chlamydomonas reinhardtii* (Chlamydomonas throughout) is a superior model organism to study important cellular processes in photosynthetic eukaryotes, including but not limited to: photosynthesis [1], cell cycle [2], lipid accumulation [3], stress responses [4,5], and biofuel production [6]. Chlamydomonas has several advantages that can be leveraged: (1) Vegetative cells are haploid, therefore mutant phenotypes present easily; (2) Cells proliferate quickly with 6~8 h doubling time under normal growth conditions; (3) Cells grow in light using photosynthesis but also in the dark with a supplied carbon source, allowing for maintenance of photosynthetic mutants in the dark; (4) All three genomes (nucleus, chloroplast, mitochondrion) of Chlamydomonas are sequenced and transformable, making it an excellent model to study inter-organellar communications [7,8,9]; (5) It has generally smaller gene families than land plants, simplifying genetic and functional analyses [8,10]; and (6) Many cellular processes in Chlamydomonas have similarities with land plants, making it an excellent model organism to identify novel genes/pathways with potential applications in crops.

Chlamydomonas has rich genetic and genomic resources, and its unicellular nature enables high-throughput approaches and functional genomics. Many transcriptomic analyses under various conditions exist in Chlamydomonas [11], including light stress [12], nutrient limitation [13,14,15], metal deficiency [16,17], day/night cycle [18,19], oxidative stress [20], and temperature stresses [21,22,23]. Additionally, several proteomics datasets for specific cellular structures are available, including chloroplasts [24], pyrenoids [25,26], mitochondria [27], and cilia [28,29]. Two powerful and efficient gene cloning approaches are well-established in Chlamydomonas: the Modular Cloning (MoClo) Toolkit (Golden Gate cloning kit for synthetic biology) [30] and the recombineering pipeline to clone large and complex genes [31]. Additionally, highly efficient insertional mutagenesis and CRISPR gene editing tools are well-optimized in Chlamydomonas [32,33,34,35,36]. Furthermore, a genome-saturating, indexed, mapped Chlamydomonas mutant library is available for both reverse and forward genetic screens (Chlamydomonas Library Project, CLiP, https://www.chlamylibrary.org/, accessed on 10 January 2022) [37,38]. The mutant library also allows for high-throughput, quantitative phenotyping of genome-wide mutants in pooled cultures under different conditions [37,38]. The mutant library is maintained and distributed by the Chlamydomonas Resource Center at the University of Minnesota, which also has several other collections of mutants in Chlamydomonas (https://www.chlamycollection.org/, accessed on 10 January 2022). These resources and tools accelerate research using Chlamydomonas as a model organism.

Two commonly used Chlamydomonas background strains are CC-1690 and CC-5325. CC in the strain names is short for Chlamydomonas Resource Center. CC-1690 is also called 21gr or Sager 21gr because it was from Dr. Ruth Sager at the Sidney Farber Cancer Institute in 1983 [39,40]. CC-1690 is mating type plus (mt^+^); it has been used for most of the cDNA libraries, subsequent Expressed Sequence Tags (ESTs), and the Chlamydomonas Genome Project [41]. Its genome has been sequenced multiple times by several laboratories [42,43]. It has also been used for several transcriptome analyses [12,23] and is an often-preferred background strain for photosynthesis- or stress-related experiments [25,44,45,46]. CC-5325 is relatively new, available to the Chlamydomonas community since 2014 [47], but has been frequently used because it is the background strain of the Chlamydomonas CLiP mutant Library [37,38]. CC-5325 is mating type minus (mt^−^) and it is the same as CC-4533, which was from an independent cryogenic storage copy. CC-5325 is also called CMJ030 (the 30th select Chlamydomonas strain from the laboratory of Dr. Martin Jonikas). CC-5325 was isolated from a cross between two parental strains: 4A^−^ (cell wall intact) and D66^+^ (cell wall deficient). CC-5325 was chosen as the background strain of the CLiP mutant library because of its several desirable features [37,47]: it recovers well from cryogenic storage, has high electroporation and mating efficiencies, grows efficiently in heterotrophic and photoautotrophic conditions, has a functional carbon concentrating mechanism (CCM) and low clumping in liquid culture, swims normally, and has minimal adherence to glass. Since 2015, more than 500 scientific papers have been published using CC-5325 or mutants in its background, for example: [25,48,49,50,51,52,53]. Genome-wide mutant screens have been performed under diverse growth conditions by using the CLiP mutant library [38,54].

Despite its widespread use, some questions regarding CC-5325 remain. Although a genome-wide screen for mutants deficient in photosynthesis was performed by using the CLiP mutant library [38], detailed analysis of photosynthesis in CC-5325, the CLiP background, is rare. Chaux et al. reported some photosynthetic measurements in CC-5325 when characterizing a CLiP mutant with deficiency in the light reaction of photosynthesis [48]. However, photosynthesis in CC-5325 has not been thoroughly evaluated by comparing with other commonly used laboratory strains, such as CC-1690. This information will be helpful for employing the CLiP mutant library to study mutants deficient in photosynthesis. Additionally, because one parent strain of CC-5325, D66^+^, is cell-wall deficient, CC-5325 is often referred to as a cell-wall deficient strain and the Chlamydomonas Resource Center labels it with *cw15* for cell wall deficiency. The name of *cw15* comes from a cell-wall deficient mutant, in which the cell wall is absent or greatly reduced as compared to wildtype [55,56]. Consequently, to eliminate the effects of cell wall deficiency, researchers often cross CLiP mutants in the CC-5325 background to other cell-wall intact strains for further analysis. However, the integrity of the cell wall in CC-5325 has not been carefully investigated. Furthermore, among the 121 different conditions for genome-wide mutant screens using the Chlamydomonas CLiP mutant library, screens related to circadian rhythms are lacking [54]. It is unclear if CC-5325 displays normal circadian rhythms and how well it can maintain its circadian rhythms.

To address these questions mentioned above, we performed comparative phenotypic analyses in CC-1690 and CC-5325. Besides CC-1690 and CC-5325, there are also some other commonly used background strains, e.g., CC-124 and CC-125 [42,43]. We chose to compare CC-1690 and CC-5325 because of our interest in photosynthesis and heat responses, the available transcriptomes of CC-1690 in responses to light and high temperatures [12,23], and the genome-wide screens for mutants deficient in photosynthesis and heat responses by using the CLiP mutant library [38,54]. In this work, we focused on specific phenotypes related to photosynthesis, thermotolerance, circadian rhythm of phototaxis, and ciliary features in these two strains.

Our results showed that: (1) CC-5325 had lower photosynthetic efficiency than CC-1690, but it grew better than CC-1690 in dark conditions, suggesting CC-5325 is a suitable background strain for mutants deficient in photosynthesis; (2) CC-5325 was more heat-sensitive than CC-1690; (3) CC-5325 had an intact cell wall with comparable integrity to that in CC-1690, except that the cell wall in CC-5325 appeared thinner than that in CC-1690; (4) CC-1690 could maintain a robust circadian rhythm of phototaxis in either constant darkness or constant light, but CC-5325 could not; (5) CC-5325 had longer cilia in the medium with acetate but slower swimming speed in the medium without nitrogen and acetate than CC-1690. Our results will be useful for the Chlamydomonas community to better understand the differences between these two commonly used background strains, especially CC-5325 as the background of the CLiP mutant library.

## 2. Results

We grew Chlamydomonas strains CC-1690 and CC-5325 with tris-acetate-phosphate (TAP) medium in photobioreactors (PBRs) under well-controlled conditions. Under the same cultivation conditions in PBRs, these two strains had different cell shapes and sizes (Figure 1). CC-5325 had larger cell volume than CC-1690. Despite these morphological differences, both had comparable growth rates, as estimated by the rate of increased OD_680_ or chlorophyll content (Figure 2).

We next evaluated pigment content using algal cultures of CC-1690 and CC-5325 grown under well-controlled conditions in PBRs as mentioned above. CC-5325 had more chlorophyll and carotenoids per cell than CC-1690, but the differences were mainly due to the larger cell volume of CC-5325, as these two strains had no differences in pigments per cell volume (Figure 3a–d). Both strains had the same chlorophyll a/b ratio (Figure 3e). However, CC-1690 had a higher chlorophyll/carotenoid ratio than CC-5325, suggesting more carotenoids in CC-5325 than CC-1690 (Figure 3f).

To evaluate photosynthetic performance, we imaged room temperature chlorophyll fluorescence in algal spots grown on TAP plates using FluorCam, a chlorophyll fluorescence imager (Photon System Instruments). Both strains had indistinguishable PSII efficiency in dark and light on plates. However, CC-5325 had higher non-photochemical quenching (NPQ) than CC-1690 (Figure 4a). NPQ is one of the most important photoprotective pathways in photosynthetic organisms, helping dissipate excess light energy and protecting photosynthesis [57,58]. To investigate photosynthetic parameters further, we used liquid algal cultures grown in PBRs for detailed photosynthetic measurements. We performed 77 K chlorophyll fluorescence to check the relative antenna size of PSII and PSI [59]. CC-5325 had smaller PSII antenna size than CC-1690 (Figure 4b,c). By using a custom-designed kinetic spectrophotometer/fluorometer, we showed that CC-5325 had higher PSII maximum efficiency in dark-adapted cells but lower PSII operating efficiency under 400 μmol photons m^−2^ s^−1^ light than CC-1690 (Figure 4d). In response to the same light intensities, both strains had little difference in Q_A_ redox status, except at 200 μmol photons m^−2^ s^−1^ light (Figure 4e). By using the PSII% (relative PSII antenna size or percentage of light distributed to PSII) estimated from our 77 K chlorophyll fluorescence data and PSII efficiency measured by room temperature chlorophyll fluorescence, we quantified rates of linear electron flow (LEF) in response to light in these two strains (Figure 4f). CC-1690 had higher LEF than CC-5325 only at 400 μmol photons m^−2^ s^−1^ light. However, CC-5325 had much higher NPQ than CC-1690 under all three light intensities tested (Figure 4g). Additionally, by using a Hansatech Chlorolab 2 Clark-type oxygen electrode, we found CC-1690 had higher gross O_2_ evolution rates than CC-5325 in response to the same light intensities while both had no differences in respiration rates (Figure 4h,i). Our results suggested that CC-5325 had lower photosynthetic efficiency than CC-1690.

We harvested algal cultures of CC-1690 and CC-5325 from PBRs, spotted them on agar plates with either TAP (with acetate, the supplied carbon source) or TP (Tris-phosphate medium, without acetate) medium, and grew them under different light and temperature conditions in incubators (Figure 5). CC-5325 grew better than CC-1690 in the dark with acetate (Figure 5a). CC-1690 could grow in the dark but appeared to have slowed growth as compared to CC-5325. At 25 °C and 35 °C under 150 μmol photons m^−2^ s^−1^ light in incubators, the growth of CC-1690 and CC-5325 was indistinguishable (Figure 5b–e,g). To evaluate the thermotolerance of these two strains under well-controlled temperature conditions in liquid cultures, we exposed the algal cultures to 43 °C for 2 h in PBRs, then spotted the cells on agar plates to measure viability. CC-5325 barely survived the heat of 43 °C, while CC-1690 could survive 43 °C heat better than CC-5325 (Figure 5f). Our results showed that both strains grew well photoautotropically and mixotrophically, but CC-5325 grew better under heterotrophic conditions while CC-1690 had higher thermotolerance than CC-5325.

Thermotolerance is linked to cell wall properties [60,61], although how cell walls affect thermotolerance is unclear. Because CC-5325 is a progeny from a cross between 4A^−^ (cell wall intact) and D66^+^ (cell wall deficient), CC-5325 is often thought to be cell wall deficient. We evaluated the cell walls of CC-1690 and CC-5325 based on freezing tolerance, a detergent assay, and cell wall imaging (Figure 6, Figure 7 and Figure 8). Cell-wall deficient strains are thought to recover better from liquid nitrogen freezing than cell-wall intact strains [62]. Both CC-1690 and CC-5325 recovered equally well from liquid nitrogen freezing (Figure 6). Next, we used a detergent assay to evaluate the integrity of the cell wall. Triton X-100 is a detergent that disrupts lipid-based membranes and causes cells without an intact cell wall to lyse. Strains without an intact cell wall or reduced cell wall integrity will have more damaged cells in the presence of Triton X-100, thus releasing more chlorophyll into the supernatant, which could be quantified using absorbance at 435 nm. We included the well-studied cell-wall-deficient line *cw15* as a control for this assay [55,56]. Without Triton X-100 treatment, there was little chlorophyll released in all three lines (Figure 7a). With Triton X-100 treatment, all three lines had increased chlorophyll release, but CC-5325 had the least chlorophyll release while *cw15* had the most, demonstrating the effectiveness of our assay. By using a Coulter Counter, we quantified the number of intact cells with and without Triton X-100 treatment (Figure 7b). While Triton X-100 did not affect CC-1690 or CC-5325, it damaged nearly all cells of *cw15*. Our freezing and detergent assays showed that CC-1690 and CC-5325 had comparable cell wall integrity.

To directly visualize and quantify the cell wall, we used the lectin Concanavalin A as a cell wall stain and Syto 13^TM^ Green Fluorescent Nucleic Acid to label the nucleus, then imaged the cells using a ZEISS Elyra 7 super-resolution microscope (Figure 8). Our results showed that CC-1690 and CC-5325 had continuous and intact cell walls, while *cw15* has discontinuous and broken cell walls, as expected for a cell-wall deficient mutant (Figure 8a–f). We calculated the total cell wall fluorescence intensity along each cell and normalized it to the cell perimeter length. The mean cell wall fluorescence intensity of CC-5325 was 84% of that in CC-1690, while that in *cw15* was 17% relative to CC-1690 (Figure 8g). Based on our results of freezing recovery, detergent assays, and cell wall imaging, we conclude that CC-5325 is not as cell-wall deficient as previously described. Instead, CC-5325 has an intact cell wall with comparable integrity to CC-1690, but the cell wall in CC-5325 is possibly thinner than that of CC-1690.

Next, we investigated the circadian rhythm of phototaxis in CC-1690 and CC-5325 because both phototaxis and circadian rhythms are important for photosynthesis, thermotolerance, and cell growth [63,64,65,66]. Rhythmic phototaxis (swimming towards light) in constant light or darkness is a known circadian-regulated activity in Chlamydomonas and can serve as a read-out of clock function [67,68]. Circadian phototaxis was determined in day/night entrained cultures followed by either “constant darkness” or “constant light” free-running conditions (Figure 9a,b). Under constant dark conditions (Figure 9c–f), the darkness was interrupted every hour by a narrow, weak light beam directed through the culture for 15 min to allow for phototaxis measurements. In constant light conditions (Figure 9g–j), the white background light was turned off every hour for the 15 min of phototaxis measurement via the weak light beam. CC-1690 showed a robust circadian rhythm of phototaxis in both constant darkness and constant light. The period of the rhythm for CC-1690 was 27.65 ± 0.13 h in constant darkness (n = 22, mean ± SE) and 23.61 ± 0.16 h in constant light (n = 12, mean ± SE). However, in CC-5325 a sustained circadian rhythm of phototaxis could neither be detected in constant darkness nor in constant light. This inability was not simply due to an inability of the strain to undergo phototaxis (Figure 9d,h). When monitoring the phototaxis of CC-5325, there seemed a start of a rhythm in nearly all cultures tested in constant light and in most of the cultures tested in constant darkness but only for the first 12 to 36 h of constant conditions. The lack of a sustained circadian rhythm of phototaxis in CC-5325 was independent of cell densities (from 0.5 × 10^6^ to 1.6 × 10^6^ cells mL^−1^). Our results clearly demonstrated that CC-1690 could maintain a sustained circadian rhythm of phototaxis but CC-5325 could not.

Because cilia affect phototaxis, we further investigated cilia morphology and swimming speed. Cells were grown in either TAP (with acetate) or M-N (without acetate and nitrogen) medium with constant agitation and light. The M-N condition induces gamete formation and eliminates the effect of the cell cycle on percent ciliation and ciliary length. The percent of cells with cilia was comparable between CC-1690 and CC-5325 in both TAP and M-N medium (Figure 10a,b). However, CC-5325 had significantly longer cilia than CC-1690 in TAP medium; this trend might be also true for cells in M-N medium, but the differences were not significant (Figure 10c,d). We quantified swimming speed by creating traces of how far the cells moved during a 1 s exposure using a darkfield microscopy (Figure 10e,f). CC-1690 and CC-5325 had a similar swimming speed when grown in TAP. When grown in M-N, CC-1690 swam faster than CC-5325 (Figure 10g,h).

## 3. Discussion

We compared phenotypic differences between CC-1690 and CC-5325 by focusing on characteristics that are important for algal growth, including cell size, growth in liquid and on agar plates, photosynthesis, thermotolerance, cell wall integrity, circadian rhythm of phototaxis, ciliary percentage/length, and swimming speed. These two frequently used background strains displayed notable differences in these characteristics (Figure 11).

Both strains grew well in liquid medium and on agar plates under normal conditions (Figure 2 and Figure 5). CC-5325 grew more robustly in dark with acetate than CC-1690 (Figure 5a), making it a promising background strain for mutants deficient in photosynthesis, which are often maintained in dark conditions with supplied carbon source. The reduced photosynthetic efficiency in CC-5325 may be related to smaller PSII antenna size and/or increased NPQ as compared to CC-1690 (Figure 4c,g,h). The NPQ differences in these two strains may be relative, since NPQ from different strains may not be compared directly [69,70]. The increased NPQ in CC-5325 may be related to the reduced chlorophyll/carotenoid ratio (more carotenoids) (Figure 3f). Carotenoids have reported roles in photoprotection and one of the carotenoid pigments, zeaxanthin, has major roles in NPQ formation [4,71]. The bigger PSI peak of 77 K chlorophyll fluorescence in CC-5325 (Figure 4b,c) suggested it had larger PSI antenna size but reduced PSII antenna size as compared to CC-1690 [59]. CC-5325 has lower oxygen evolution rates than CC-1690 only under light intensity equal to or greater 200 μmol photons m^−2^ s^−1^ light (Figure 4h), which may explain the comparable growth rates of these two strains in PBRs under 100 μmol photons m^−2^ s^−1^ light (Figure 2). Although CC-5325 had lower photosynthetic efficiency than CC-1690, it does not compromise its suitability as a background strain to screen for mutants deficient in photosynthesis. Because photosynthetic efficiencies are relative, it is not necessary for a background strain to have the highest photosynthetic efficiency for generating and characterizing photosynthetic mutants; normal photosynthesis would be adequate. Previous research showed that CC-5325 has a functional carbon concentrating mechanism (CCM) and several CLiP mutants deficient in CCM have been identified by using CC-5325 as a background strain [25,49,72]. CLiP mutants deficient in photosynthetic light reactions were also identified and published [48,73]. Additionally, a genome-wide mutant screen has been performed to identify genes required for photosynthesis using the CLiP mutant library [38]. Based on our results and published literature, we believe CC-5325 is a suitable background strain to study photosynthesis in Chlamydomonas.

CC-5325 has been frequently reported as a cell-wall deficient line, like *cw15* [37,50]. Our results from freezing and detergent assays showed that the cell wall integrity in CC-5325 was comparable to that in CC-1690 (Figure 6 and Figure 7). Cell wall imaging and quantification showed that CC-5325 had an intact cell wall, albeit possibly thinner cell wall than that in CC-1690 (Figure 8). The vegetative cell wall of Chlamydomonas is mainly comprised of proteins [74,75]. It is possible that these two strains have different quantities of cell wall proteins. There are three classifications of cell-wall-deficient mutants: (A) normal quality of cell walls that are unattached to the plasma membrane; (B) normal quality of cell walls that are attached to the plasma membrane; (C) minimal amount of cell walls [55]. The well-studied cell-wall-deficient mutant, *cw15*, belongs to classification C [55,75]. Our results showed that CC-5325 was clearly not a C-type cell-wall mutant. Chlamydomonas cell walls protect cells from environmental stresses [75], but the underlying mechanisms are unclear. Further investigation is needed to investigate if the reduced cell wall in CC-5325 contributes to some of its increased heat-sensitivity in comparison to CC-1690 (Figure 5f). Nevertheless, the cell wall integrity in CC-5325 is clearly much better than previously thought and its increased heat-sensitivity as compared to CC-1690 does not affect its use for studying heat responses. The CLiP mutant library has been used to screen for heat-sensitive mutants [54]. It may not be necessary to cross CLiP mutants on the background of CC-5325 to other cell-wall intact strains. Additionally, for research related to heat stress, it may generate conflicting results if the heat-sensitive CLiP mutants are crossed to other non-isogeneic lines (e.g., CC-1690) because different background strains may have different heat sensitivities, as demonstrated in our research. If crossing is needed, a potentially better option would be to cross CLiP mutants to the isogenic strain of CC-5325, which is CC-5155 (mt^+^). CC-5155 was generated by crossing CC-5325 (mt^−^) and CC-125 (mt^+^, 137c) and then backcrossing a mt^+^ progeny to CC-5325 five times [37].

The most surprising result may be the lack of a sustained circadian rhythm of phototaxis in CC-5325, although the strain can perform phototaxis (Figure 9). Our results clearly showed a sustained circadian rhythm of phototaxis in CC-1690, but CC-5325 only showed a brief rhythm of phototaxis in constant conditions for the first 12 to 36 h. This lack of a sustained circadian rhythm of phototaxis in CC-5325 was present under both constant light and constant darkness. A number of different circadian rhythms have been studied in Chlamydomonas [63,64], including phototaxis, chemotaxis, cell division, UV sensitivities, and others. How well Chlamydomonas can maintain these rhythms under various environmental conditions is less known. From our data, we cannot distinguish whether the lack of a sustained rhythm in CC-5325 is due to defects in the circadian oscillator mechanism, and thereby also affecting other circadian rhythms, or whether it is specifically due to the ability of the clock to regulate specific outputs such as phototaxis. If the former is true, the reduced photosynthetic efficiency and thermotolerance observed in CC-5325 may be related to its deficiency in maintaining circadian rhythms, as normal circadian rhythms are important for photosynthetic organisms to optimize photosynthesis and acclimate to stressful environments [65,66,76,77]. In CC-1690, constant darkness or light affected the period of the rhythm, with a longer period in constant darkness (27.65 h) than constant light (23.61 h). Differences in period length under different constant conditions have been observed in many organisms. For example, it was recently reported that the duckweed, *Lemna gibba*, showed an average period under conditions of constant darkness that was about 7 h longer than it was in constant light [78]. Transcriptome analysis in cultures under synchronizing versus free-running conditions may provide some insights about the mechanisms of the maintenance of circadian rhythms and about the basis for the observed differences between these two strains.

CC-1690 and CC-5325 also had some differences in cilia length and swimming speed (Figure 10). CC-5325 had longer cilia in TAP medium than CC-1690, consistent with the increased cell volume of CC-5325 in TAP as compared to CC-1690 (Figure 1), as ciliary length scales with cell size [79]. CC-5325 had slower swimming speed in M-N medium than CC-1690, which may be related to additional influences by positive gravitaxis (downward swimming in response to gravity), as we observed that CC-5325 gravitaxed and settled faster than CC-1690 in both TAP and M-N medium. This may be due to shape-based mechanical orientation of cells [80] rather than due to the large size of CC-5325 as another large-size strain, CC-125, does not exhibit positive gravitaxis. Additionally, the tendency to swim downward did not lead to increased clumping in CC-5325 in our algal cultivation in flasks or algal photobioreactors.

The underlying genetic causes of these phenotypic differences are of great interest. Comparison between genome sequences of CC-1690 and CC-5325 showed that these two strains had 117,059 small variants (include single nucleotide substitutions and small insertions and deletions) and 355 high impact small variants (that are predicted to cause a frameshift or premature translation stop) [37], providing potential candidates for phenotyping differences. To understand the genetic factors underlying the phenotypic differences observed between CC-1690 and CC-5325, we investigated the functional annotations of the nonsynonymous mutations in coding regions of CC-5325 (Appendix A). Li et al., 2016 reported the high confidence genomic differences that are unique to CC-5325 as compared to several other commonly used laboratory strains, including CC-1690. Among these, we identified the nonsynonymous genetic difference in coding regions of 40 genes in CC-5325, including codon changes, codon insertions/deletions, and frameshift mutations (Appendix A). Of the 40 genes containing nonsynonymous mutations, six have putative roles in regulation including three transcription factors, three transcriptional regulators, and a histidine kinase response regulator. About half of these genes have no or little function annotation. Notably, we did not observe nonsynonymous mutations in the coding regions of genes with known roles involved in photosynthesis, heat responses, cell wall biogenesis, circadian regulation, phototaxis, and cilia biology. Because CC-5325 had reduced PSII antenna size as compared to CC-1690 (Figure 4b,c), we also checked genes with known roles in state transition in Chlamydomonas. State transition is a process that reorganizes light-harvesting antenna systems and redistributes the light excitation energy between PSII and PSI to balance the energy input/consumption between the two photosystems [81]. The kinase STATE TRANSITION7 (STT7) in Chlamydomonas phosphorylates components of light-harvesting antenna complex II (LHCII), triggering the state transition [81]. Two phosphatases reverse the LHCII phosphorylation and the state transition: PPH and PBCP (Phosphoprotein Phosphatase 2C-related and PSII Core Phosphatase) [73]. However, none of these genes with known roles in state transition had nonsynonymous mutations in CC-5325. These results suggest that the observed phenotypic differences between the two background strains may be due to changes at the regulatory level rather than codon changes in the proteins with known functions involved in these biological processes. Further detailed analysis, such as quantitative trait locus (QTL) mapping, could be performed in the future to identify the causative genetic loci underpinning the phenotype differences observed between CC-1690 and CC-5325.

QTL mapping has been used frequently to identify causative genes related to a phenotype in land plants [82,83,84,85]. Such analysis is much less developed in Chlamydomonas. However, Chlamydomonas has several advantages for QTL mapping, e.g., high mating efficiency and easy mating procedure [86], fast growth on plates or in liquid, low cost in whole-genome sequencing [87,88], and abundant RNA-Seq datasets [11]. One round of mating in Chlamydomonas only takes about 2–3 weeks, from the mating process to identifiable progenies. Due to its unicellular nature, high throughput phenotyping tools can be applied to characterize progenies of Chlamydomonas efficiently, e.g., whole plate chlorophyll fluorescence imaging [89,90,91], growth quantification on plates using ImageJ [23], and circadian rhythm monitoring using an automated phototaxis machine [92]. The CLiP mutant library allows for both forward and reserve genetic screens and candidate genes identified by QTL in Chlamydomonas can be validated using available mutants in the CLiP mutant library [37]. If mutants deficient in some specific genes are unavailable in the CLiP mutant library or other mutant collections at the Chlamydomonas Resource Center, mutants disrupted in genes of interest could be generated by CRISPR-Cas9 approaches with high efficiency [34,35,36]. Recently, research conducted to phenotype and select progenies from a cross between two Chlamydomonas parental strains under different environmental conditions revealed possible selection-enriched genomic loci for improved stress adaptation and photosynthetic efficiency [93], paving the way for QTL mapping in Chlamydomonas.

## 4. Conclusions

We presented several distinct phenotypes between two commonly used laboratory background Chlamydomonas strains: CC-1690 (21gr) and CC-5325 (CLiP library background). Although CC-5325 has lower photosynthetic efficiency and thermotolerance than CC-1690 and these two strains have some differences in cilia, it does not compromise the use of CC-5325 and the CLiP mutant library to study and screen for mutants deficient in photosynthesis, heat responses, or cilia biology. Our results showed that CC-5325 had an intact cell wall with comparable integrity as compared to CC-1690, which will help reduce researchers’ concerns or confusion about its cell wall deficiency. The lack of a sustained circadian rhythm of phototaxis in CC-5325 does limit the use of the CLiP library to screen for mutants deficient in circadian rhythms. There may be no background strains that are perfect for all biological processes, but it is important to understand the phenotypic behaviors of one background strain when using it. Our research will help fill the knowledge gaps about CC-5325 and facilitate the use of the CLiP mutant library generated in this background. Finally, the rich genomic and genetic resources in Chlamydomonas will enable identification of responsible genes for phenotypes related to cell size, photosynthesis, thermotolerance, circadian rhythms, and cilia biology.

## 5. Materials and Methods

### 5.1. Strains and Culture Conditions

Chlamydomonas wildtype strain CC-1690 (also called 21gr, mt^+^) and CC-5325 (also called CMJ030, identical to CC-4533, mt^−^, CLiP library background) were ordered from the Chlamydomonas Resource Center (University of Minnesota). Algal cultures of these two strains were maintained in photobioreactors (PBRs) as described before [23]. The normal growth condition in PBRs was: 25 °C, Tris-acetate-phosphate (TAP) medium, constant 100 μmol photons m^−2^ s^−1^ light (50% red: 50% blue), constant bubbling with filtered air (0.04% CO_2_), and turbidostatic control by monitoring OD_680_ to supply fresh medium frequently and maintain cell density around 2 × 10^6^ cells mL^−1^. Under the normal growth condition at 25 °C, OD_680_ is linearly proportional to both chlorophyll content and cell density. Fresh medium was added to the culture automatically by a peristaltic pump when the OD_680_ reached the defined maximum value to dilute the culture; the pump was stopped when the OD_680_ dropped to the defined minimum value. Algal cultures then grew up at approximately exponential rate to the defined maximum OD_680_ value before the next dilution cycle. Our OD_680_ range was sufficiently small that we expected minimal nutrient limitation during our experiment. Thus, the turbidostatic mode precisely controlled the growth condition. We calculated the doubling time or relative growth rates based on the exponential growth phase when the peristaltic pump was off. The doubling time and relative growth rates we referred to were based on the increase of OD_680_ or the total chlorophyll per mL culture. The growth rates measured by OD_680_ were consistent with medium consumption rates and cell number increase quantified by a cell counter. For the 43 °C heat treatment in PBRs, the PBR temperature was gradually increased from 25 °C to 43 °C within 30 min, and then the PBR temperature was maintained at 43 °C for 2 h. By the end of 2 h heating at 43 °C, algal cultures were harvested for spotting test to evaluate viability.

### 5.2. Algal Spotting Test

Algal spotting test was performed as described before with minor modifications [23]. CC-1690 and CC-5325 cultures harvested from PBRs were diluted to 1 × 10^5^ cells mL^−1^ or 0.2 × 10^5^ cells mL^−1^ (1:20 or 1:100 dilution) and 10 μL aliquots of the diluted cultures (about 1000 or 200 cells) were spotted on 1.5% TAP agar plates and grown in temperature-controlled incubators under the indicated conditions. For conditions with light, 150 μmol photons m^−2^ s^−1^ light was provided by white LED lights with indicated intensity. After 44-h growth under light in incubators, algal spots were imaged by a dissecting Leica microscope. The whole plates with algal spots were imaged again using a regular camera after 3-day-growth for visual representations. Due to the slow growth in darkness, the algal spots and the whole plates were images after 5 days and 10 days, respectively.

### 5.3. Pigment Measurements

Pigments were quantified as described before [23]. Three biological replicates of 1 mL of PBR cultures (around 2 × 10^6^ cells mL^−1^) grown under normal condition of 25 °C were harvested, mixed with 2.5 μL of 2% Tween20 (Sigma, P9416-100ML) to help cell pelleting, centrifuged at 18,407× *g* at 4 °C to remove supernatant. Cell pellets were resuspended in 1 mL of HPLC grade methanol (100%, Sigma, 34860-4L-R), vortexed for 1 min, incubated in the dark for 5 min at 4 °C, and centrifuged at 15,000× *g* at 4 °C for 5 min. Top supernatant containing pigments was analyzed at 470, 652, 665 nm in a spectrophotometer (IMPLEN Nonophotometer P300) for carotenoids and chlorophyll a/b concentrations in μg mL^−1^ using the following equations: Chl a + Chl b = 22.12xA_652_ + 2.71xA_665_, Chl a = 16.29xA_665_ − 8.54xA_652_, and Chl b = 30.66xA_652_ − 13.58xA_665_ [94], and carotenoids = (1000xA_470_ − 2.86xChl a − 129.2xChl b)/221 [95].

### 5.4. Photosynthetic Measurements

CC-1690 and CC-5325 cultures grown at 25 °C in PBRs as mentioned above were used for all photosynthetic measurements. Algal cultures were spotted on TAP plates and grown under 150 μmol photons m^−2^ s^−1^ light at 25 °C in incubators for 4-day. Algal plates were dark-adapted for 20 min before chlorophyll fluorescence imaging at room temperature using FluorCam (Photo System Instruments). Photosynthetic measurements in liquid algal cultures were performed as described before [23]. 77 K chlorophyll fluorescence measurement were performed using an Ocean Optics spectrometer (Cat. No. OCEAN-HDX-XR, Ocean Insight, Seminole, FL, USA) and 430 nm excitation provided by light emitting diodes (LEDs). Spectral data were normalized to the PSII spectral maximum value at 686 nm and relative percentage of light distributed to PSII (or relative PSII antenna size, PSII%) was calculated using this formula: PSII% = normalized PSII peak/(normalized PSII peak + normalized PSI peak at 714 nm). Normalized PSII peak equals one. Room temperature chlorophyll fluorescence in liquid algal cultures were performed using a multi-wavelength kinetic spectrophotometer or fluorometer (measuring beam with 505 nm peak emission, measuring pulses of 100 μs duration) [23]. Aliquots of 2.5 mL algal cultures (around 12 μg chlorophyll) were sampled from PBRs, supplemented with 25 μL of fresh 0.5 M NaHCO_3_ in a fluorometry cuvette (C0918, Sigma-Aldrich, St. Louis, MO, USA) with constant stirring, and dark adapted with a 10-min exposure to far-red light (peak emission of 730 nm at ~35 μmol photons m^−2^ s^−1^) before chlorophyll fluorescence measurement in dark, and subsequent actinic light phases of 100, 200, 400 μmol photons m^−2^ s^−1^ light. The actinic light was provided by LED lights with maximal emission at 620 nm. Each light phase was about 7.5 min long. A saturating flash was applied at the end of each phase to get maximum chlorophyll fluorescence (F_m_ in dark or F_m_′ in light). Photosynthetic parameters were calculated as described [70]. PSII efficiency (ΦPSII) was calculated as 1 − F_o_/F_m_ and 1 − F′/F_m_′ for dark-adapted and light-adapted algal cells, respectively. F_o_ and F_m_ are minimal and maximal chlorophyll fluorescence in dark-adapted algal cells. F′ and F_m_′ are steady state and maximal chlorophyll fluorescence in light-adapted algal cells. Q_A_ redox status was calculated as 1 − q_L_ = 1 − (F_q_′/F_v_′) × (F_o_′/F′) = 1 − [(F_m_′ − F′)/(F_m_′ − F_o_′)] × (F_o_′/F′). q_L_ is the fraction of open PSII centres and F_q_′ is the photochemical quenching of fluorescence (Baker et al., 2007). Linear electron flow (LEF) was calculated using the formula: LEF = (actinic light) × (Qabs_fs_) × (PSII%) × (ΦPSII). Qabs_fs_ is fraction of absorbed light, assuming 0.8. PSII% was calculated from 77 K chlorophyll fluorescence as mentioned above. Non-photochemical quenching, NPQ, was calculated as (F_m_ − F_m_′)/F_m_′. Oxygen evolution was measured using the Hansatech Chlorolab 2 based on a Clark-type oxygen electrode at room temperature as described [23]. Two-mL of cells (around 10 μg chlorophyll) supplemented with 20 μL of 0.5 M NaHCO_3_ were incubated in the dark for 10 min with stirring before the light phases of 100, 200, 400 μmol photons m^−2^ s^−1^ light. Each light lasted 5 min followed by 2 min dark. The rates of oxygen evolution and respiration were measured at the end of each light and dark phase, respectively.

### 5.5. Algal Cryogenic Freezing and Thawing

CC-1690 and CC-5325 cultures around a cell density of 2 × 10^6^ cells mL^−1^ were harvested from PBRs grown at 25 °C and frozen in 10% cryoprotective agent (CPA, 1:10 volume ratio of 100% methanol and fresh TAP medium). Algal cultures of 450 μL was added to 450 μL of 10% CPA (final CPA concentration was 5%) in cryo-tubes, mixed gently, transferred to a CoolCell foam freezer container, incubated in a −80 °C freezer for 4 h, and finally stored in a liquid nitrogen freezer. During the thawing process, cryo-tubes with algal samples were thawed in a 37 °C water bath for 5 min and centrifuged at 1000× *g* for 5 min at room temperature to remove the supernatant. Fresh TAP medium of 900 μL was added to resuspend cell pellets, followed by dark storage of the cryo-tubes without agitation at room temperature overnight. The next day, the cryo-tubes were centrifuged again at 1000× *g* for 5 min at room temperature to remove the supernatant. Cell pellets were resuspended with 200 μL of fresh TAP medium and 10 μL of the resuspended algal cultures were used for the spotting test to evaluate viability as mentioned above.

### 5.6. Detergent Assay for Cell Wall Integrity

CC-1690 and CC-5325 cultures around a cell density of 2 × 10^6^ cells mL^−1^ were harvested from PBRs grown at 25 °C as mentioned above. A cell-wall deficient strain, *cw15* [55], was received from the Umen laboratory at the Donald Danforth Plant Science Center and used as a control. *cw15* could not grow well in PBRs probably because its cell wall deficiency made it sensitive to the constant air bubbling in PBRs. So shaker cultures of *cw15* around the similar cell density were used for the detergent assay. Growth condition of *cw15* on the shaker was 25 °C, around 100 μmol photons m^−2^ s^−1^ light, and in TAP medium. One-mL algal cultures were used for chlorophyll quantification [94] and another 1-mL cultures (with 0.05% Triton X-100 or with TAP medium) were immediately vortexed vigorously for 30 s then incubated in dark for 10 min. Then, 500 μL of each sample was withdrawn and centrifuged at 13,000× *g* to remove cell debris. The absorbance of the supernatant was measured at 435 nm and normalized to chlorophyll contents. The number of intact cells were measured using a Coulter Counter (Multisizer 3, Beckman Counter, Brea, CA, USA).

### 5.7. Cell Wall Staining and Quantification

Ten mL cultures of CC-1690, CC-5325, and *cw15* cells as mentioned above were collected and concentrated to 2 × 10^7^ cells mL^−1^ by centrifugation at 1500× *g* for 5 min at room temperature. Cells were stained with Concanavalin A (ConA, Alexa Fluor^®^ 594 conjugate) (100 μg mL^−1^, ThermoFisher Scientific) [96] and Syto 13 Green Fluorescent Nucleic Acid Stain (5 μM mL^−1^, ThermoFisher Scientific, Waltham, MA, USA) for 30 min at room temperature. Stained cells were washed first and then fixed in 4% paraformaldehyde (EM Science, Hatfield, PA, USA) and followed by 1X rinse in TAP medium. Fluorescence images were acquired using a 40X C-Aprochromat objective lens (numerical aperture 1.2) in apotome mode with a ZEISS Elyra 7 super-resolution microscope (ZEISS, Oberkochen, Germany). The 488 nm and 561 nm laser lines with BP 420–480 + 495–550 and BP 570–620 + LP 655 emission filters were used for Syto 13 and ConA, respectively in fast frame sequential mode at 50 ms exposure using dual pco.edge 4.2 sCMOS detectors (PCO AG, Kelheim, Germany). All two-dimensional (2D) images (with 5 phases) were acquired at 49 nm x-y pixel resolution and reconstructed using the SIM^2^ module and 2D+ processing with standard live sharpness in ZEN Black 3.0 SR FP2 software. For each strain, 100 cells were quantified by free hand drawing a line intensity profile along the cell wall. The fluorescence intensity of each cell wall periphery was normalized by its perimeter length.

### 5.8. Circadian Rhythm of Phototaxis

CC-1690 and CC-5325 were grown in 125-mL Erlenmeyer flasks containing 50-mL photoautotrophic Tris-phosphate (TP) minimum medium without acetate. Cultures were grown on a shaker at 17 to 19 °C under synchronizing 12 h light/12 h dark cycles. The light intensity during the light phase was about 20 μmol photons m^−2^ s^−1^. When cultures reached cell concentrations between 3.5 × 10^5^ and 2.3 × 10^6^ cells mL^−1^, culture samples were placed into small Petri dishes and their phototaxis was monitored at 20 °C as described [92] with the following modifications. A box around the phototaxis machine with a heating element and ventilators allowed the external temperature around the machine to be kept at a constant but slightly elevated temperature of 24 °C compared to the machine. Additionally, temperature recorders (Tempo Disc, BlueMaestro, The Woodlands, TX, USA) that fit into the Petri dish slots allowed for a more precise monitoring of the temperature the culture samples were exposed to. An algorithm as described before was used to analyze the phototaxis data for their periods [92]. Three independent experiments were conducted for each of the constant darkness and constant light conditions with 1 or 2 different cultures per strain per experiment and 3 to 4 replicate samples per culture.

### 5.9. Ciliary Measurements and Swimming Speed Assay

CC-5325 and CC-1690 cells were grown overnight with constant agitation and light at 100 μmol photons m^−2^ s^−1^ light in either liquid TAP or M-N (minimal medium without acetate and nitrogen) to induce gamete formation and eliminate the effects of the cell cycle on cilia length. A sample of each strain was fixed by mixing 1:1 with 2% glutaraldehyde for a final concentration of 1% glutaraldehyde. After fixation, cells were imaged with a DIC microscope. Images were analyzed in ImageJ/FIJI using the segmented line tool. Each cell has two cilia with the same length. One cilium per cell was traced from base to tip, then fit using the “Fit Spline” function [97]. The length of the line was then measured and converted to micrometers. Cells were only included in the measurement if both cilia were visible. The ciliation percentage was obtained by counting the number of cells with at least 1 visible cilium out of 100 cells.

For swimming speed assay, cells were allowed to settle for 15 min so that swimmers could be selected. The swimmers taken from the top of the cultures were diluted 1:10 in the same media they were grown in (either TAP or M-N). To give the cells appropriate room to swim between the slide and coverslip, a chamber was created on the slides using double stick tape. Cells were placed in the chamber and a coverslip was placed on top. Samples were imaged using a darkfield microscope set to a 1-s exposure. The resulting images show traces representing where and how far the cells swam during the 1-s exposure [98]. To estimate the swimming speed, the traces were measured in ImageJ/FIJI using the segmented line tool and the “Fit Spline” function. Superplots for ciliary length and swimming speed were created by measuring the mean of 3 biological replicates [99].

### 5.10. Genome Changes Unique to CC-5325

High confidence, unique, nonsynonymous genetic difference in coding regions of 40 genes in CC-5325 as compared to other commonly used background strains were identified using the published data in Li et al., 2016. To assess the functions of these genes, we compiled the MapMan and GO functional annotations as well as the phytozome descriptions and deflines. Putative transcription factors were identified using the Plant Transcription Factor Database (PTFDB) [100].

### 5.11. Statistical Analysis

A two-tailed *t*-test with unequal variance was used for statistical analysis. *p* > 0.05, not significant, or ns; *, 0.01 < *p* < 0.05; **, 0.001 < *p* <0.01; ***, *p* < 0.001.

## Figures and Tables

**Figure 1 plants-11-00585-f001:**
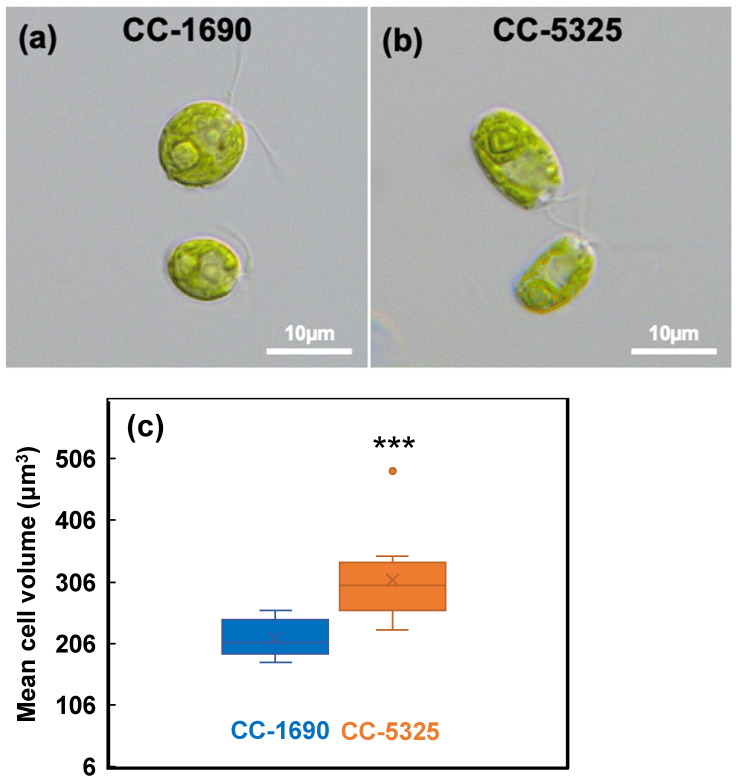
**CC-5325 had larger cell volume than CC-1690.** (**a**,**b**) Light microscopic images of Chlamydomonas cells. (**c**) Boxplot of mean cell volume determined using a Coulter Counter. N = 12 biological replicates for each strain. Statistical analyses were performed using a two-tailed *t*-test assuming unequal variance by comparing with CC-1690. ***, *p* < 0.001.

**Figure 2 plants-11-00585-f002:**
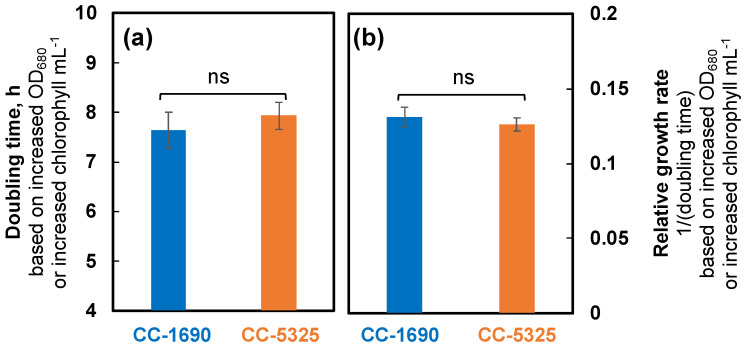
**CC-1690 and CC-5325 had comparable growth rates under mixotrophic conditions in photobioreactors (PBRs).** Chlamydomonas cells were grown in turbidostatically controlled PBRs in Tris-acetate phosphate (TAP) medium at 25 °C with a light intensity of 100 μmol photons m^−2^ s^−1^ and constantly bubbling of air. Doubling time (**a**) and relative growth rates ((**b**), inverse of doubling time) were calculated based on the exponential increase of OD_680_, which is proportional to total chlorophyll content in unit of μg chlorophyll mL^−1^ and cell density at 25 °C. Mean ± SE, N = 3 biological replicates. Statistical analyses were performed using a two-tailed *t*-test assuming unequal variance by comparing the two strains. Not significant, ns.

**Figure 3 plants-11-00585-f003:**
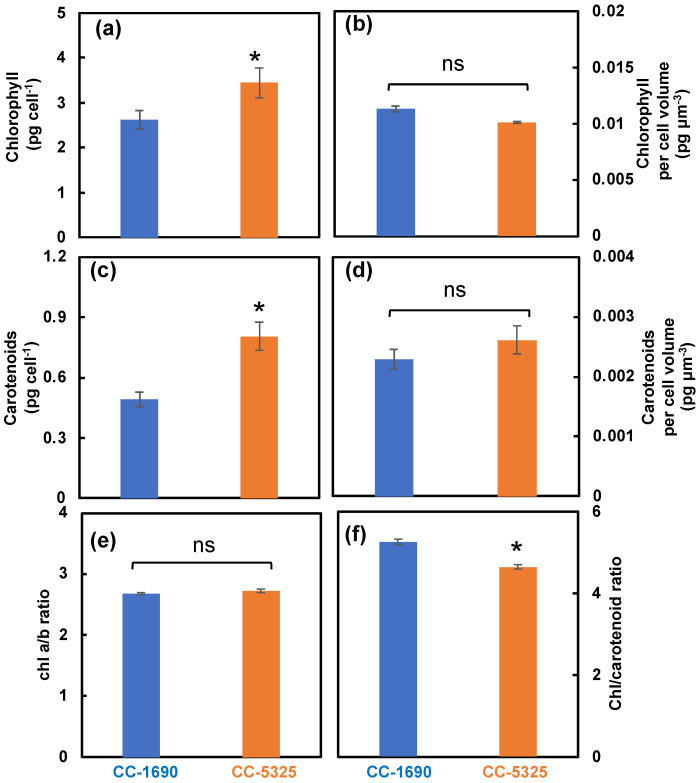
**CC-5325 had more chlorophyll and carotenoids per cell than CC-1690, mainly due to the larger cell volume of CC-5325.** Chlorophyll and carotenoids measured by absorbance change using a spectrophotometer. (**a**,**b**) Chlorophyll content per cell and per cell volume, respectively. (**c**,**d**) Carotenoid content per cell and per cell volume, respectively. (**e**) Chlorophyll a/b ratio. (**f**) Chlorophyll/carotenoid ratio. Mean ± SE, N = 3 biological replicates. Statistical analyses were performed using a two-tailed *t*-test assuming unequal variance by comparing with CC-1690: *, *p* < 0.05; ns, not significant.

**Figure 4 plants-11-00585-f004:**
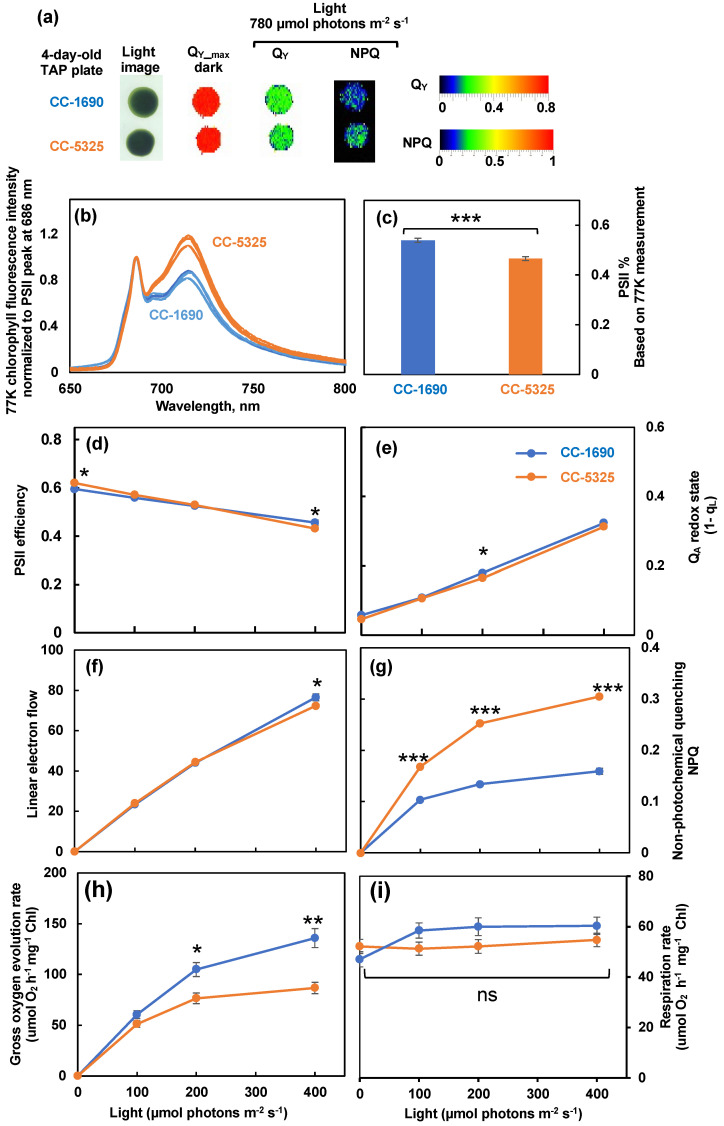
**CC-1690 had higher photosynthetic efficiency than CC-5325.** (**a**) Chlorophyll fluorescence imaging of 4-day-old algal spots grown on TAP plates at 25 °C under 150 μmol photons m^−2^ s^−1^ white LED lights in incubators. Chlorophyll fluorescence imaging was performed by using FluorCam (Photo System Instruments). Q_Y-max_, maximum PSII efficiency in dark-adapted plates. Under 780 μmol photons m^−2^ s^−1^ light, Q_Y_ and NPQ are PSII operating efficiency and non-photochemical quenching in light-adapted plates, respectively. Color bars represent values of Q_Y_ and NPQ. (**b**–**i**) Algal liquid cultures grown in PBRs under the same condition as in Figure 2 were harvested for photosynthetic measurements. (**b**) 77 K chlorophyll fluorescence normalized to PSII peak at 686 nm. (**c**) PSII%, relative PSII antenna size, or light distributed to PSII, estimated from 77 K chlorophyll fluorescence data in (**b**). (**d**–**g**) Room temperature chlorophyll fluorescence measurements were performed in algal liquid cultures using a custom-designed spectrophotometer. See methods for details. (**d**) PSII efficiency, the data at 0 μmol photons m^−2^ s^−1^ light are the maximum PSII efficiency in dark and data from light phase are PSII operating efficiency in light-adapted cells. (**e**) Q_A_ redox state, the redox state of chloroplastic quinone A (Q_A_), the primary electron acceptor downstream of PSII; the bigger number of Q_A_ redox state means more reduced Q_A_. (**f**) Linear electron flow. (**g**) Non-photochemical quenching, NPQ. (**h**,**i**) Gross O_2_ evolution rates and respiration rates, measured using a Hansatech Chlorolab 2 Clark-type oxygen electrode. Mean ± SE, N = 3 biological replicates. (**c**–**i**). All figures have error bars, but some error bars may be too small to be visible on the graph. Statistical analyses were performed using a two-tailed *t*-test assuming unequal variance by comparing with CC-1690 under the same experimental conditions. *, 0.01 < *p* < 0.05; **, 0.001 < *p* <0.01; ***, *p* < 0.001; ns, not significant. (**d**–**i**) The positions of asterisks match the corresponding light intensities.

**Figure 5 plants-11-00585-f005:**
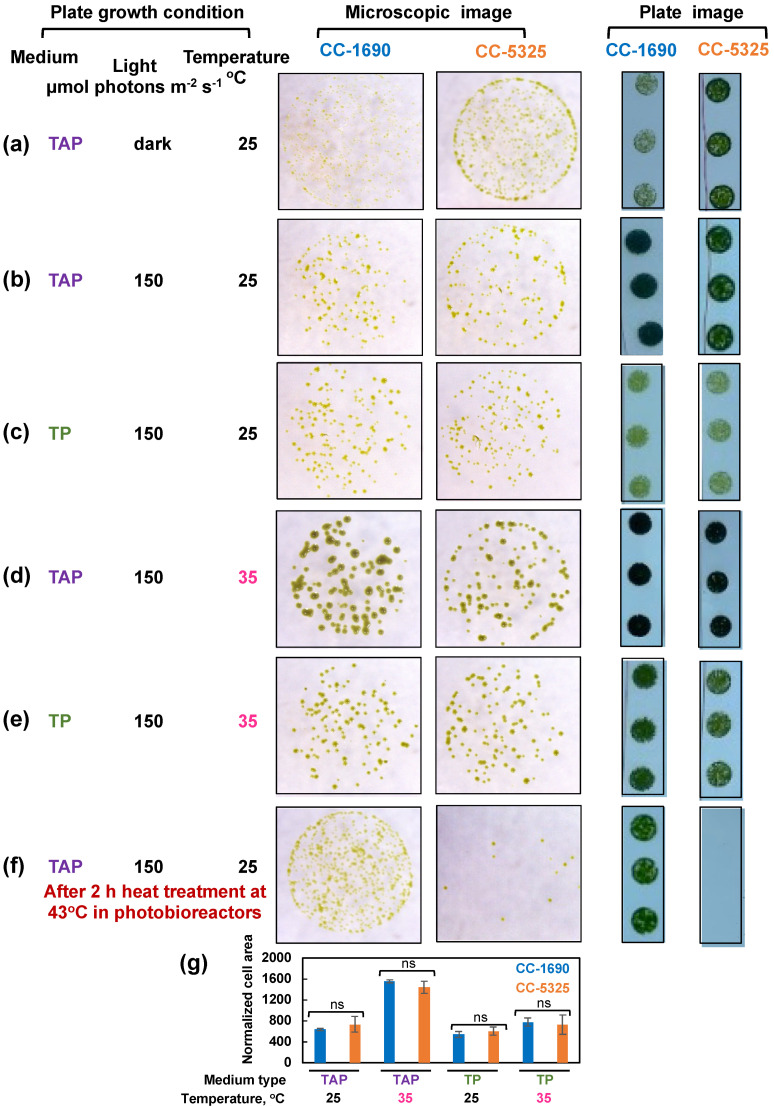
**CC-5325 grew better in the dark with acetate but was more heat-sensitive than CC-1690.** (**a**–**e**) Algal cells grown in PBRs at 25 °C under the same condition as in Figure 2 were harvested, diluted, and spotted on agar plates, and grown under the indicated growth conditions in temperature-controlled incubators. Cultures with the same cell density were used for each spot. (**f**) Algal cells were heat-treated at 43 °C for 2-h in PBRs before spotting; cultures with equal volume and the same dilution were used for each spot. TAP, Tris-acetate-phosphate medium with acetate. TP, Tris-phosphate medium without acetate. The same dilution and growth duration were used for the two strains under the same condition. (**a**,**f**) Algal cultures with 1:20 dilution, about 1000 cells in 10 μL, were used for spotting. (**b**–**e**) Algal cultures with 1:100 dilution, about 200 cells in 10 μL, were used for spotting. Single algal spots were imaged using a dissect microscope after 5-day (**a**) or 44-h (**b**–**f**) of growth. Plate images (all 1:20 dilution) were taken after 10-day (**a**) or 3-days (**b**–**f**) of growth using a regular camera. Images shown are representative of three biological replicates. (**g**) Normalized cell area. The 44-h spotting images were analyzed by ImageJ to get cell areas which were then normalized to the number of cells spotted. Statistical analyses were performed using a two-tailed *t*-test assuming unequal variance by comparing with CC-1690 under the same experimental conditions. Not significant, ns. Such quantification was not performed for (**a**,**f**) because the cell areas for one of the strains were too small to quantify.

**Figure 6 plants-11-00585-f006:**
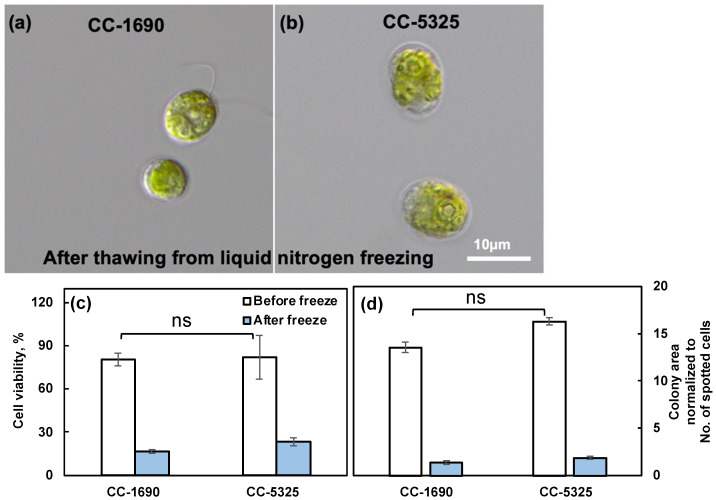
**CC-1690 and CC-5325 recovered from liquid nitrogen freezing equally well.** (**a**,**b**) Representative images of CC-1690 and CC-5325 after thawing from liquid nitrogen freezing. (**c**) Cell viability before and after freezing quantified by the No. of colonies on TAP plates. (**d**) Colony area on TAP plates for cells before and after freezing, normalized to the total number of cells spotted for each condition. The numbers of intact cells before and after freezing in liquid cultures were measured using a Coulter Counter and diluted cultures were spotted on TAP plates. TAP plates with algal spots were grown at 25 °C under 150 μmol photons m^−2^ s^−1^ white LED lights for 44-h before plate imaging. Colony numbers and areas were quantified using ImageJ. Cell viability was calculated by the No. of colonies on TAP plates divided by the No. of cells spotted. Mean ± SE, N = 3 biological replicates. Statistical analyses were performed using a two-tailed *t*-test assuming unequal variance by comparing the two strains under the same experimental conditions. No significant differences between these two strains under the same conditions; ns, not significant, *p* > 0.05.

**Figure 7 plants-11-00585-f007:**
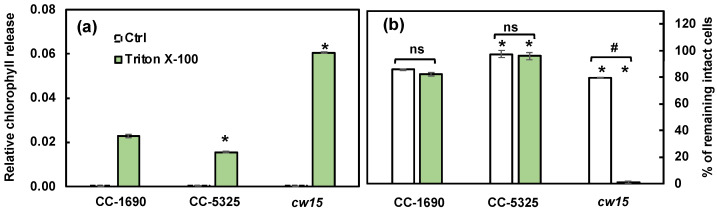
**CC-1690 and CC-5325 had comparable resistance to the detergent Triton X-100.** Detergent assay was performed to check cell wall integrity. Triton X-100 is a detergent to disrupt lipid-based membranes and cause cells without an intact cell wall to lyse. Algal samples with and without 0.05% Triton X-100 were vortexed vigorously. Strains without intact cell walls will have more broken cells in the presence of Triton X-100 and thus more released chlorophyll than strains with intact cell walls. (**a**) Released chlorophyll from damaged cells was quantified using absorbance at 435 nm. (**b**) The percentage of intact cells was quantified using a Coulter Counter. Mean ± SE, N = 3 biological replicates. Statistical analyses were performed using a two-tailed *t*-test assuming unequal variance by comparing with CC-1690 under the same experimental conditions (*, *p* < 0.05) or within the same strain in the presence or absence of Triton X-100 (#, *p* < 0.05; ns, not significant, *p* > 0.05).

**Figure 8 plants-11-00585-f008:**
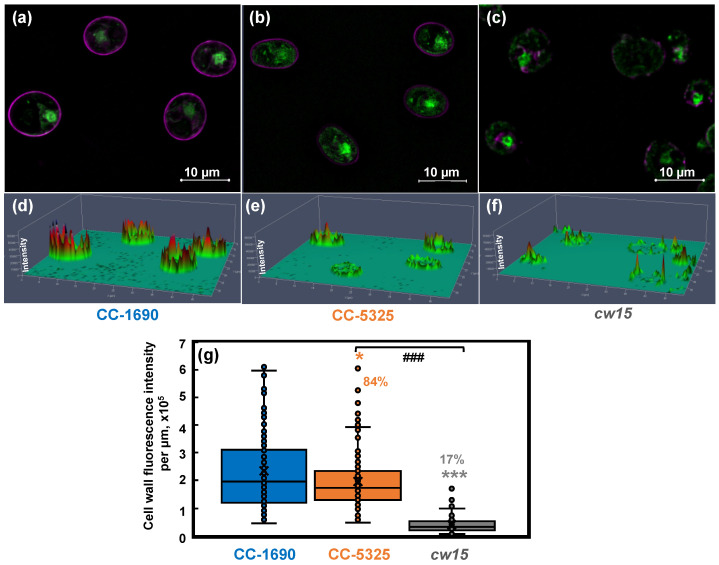
**CC-5325 had intact cell wall which may be thinner than that in CC-1690.** CC-1690, CC-5325, and *cw15* cells were stained with Concanavalin A for cell wall imaging and Syto 13^TM^ Green Fluorescent Nucleic Acid to label the nucleus. *cw15*, a cell-wall-deficient line, was used as a control. (**a**–**c**) Representative images for cell wall stain (magenta color) and nuclear stain (green color) for three different strains. (**d**–**f**) Cell wall fluorescence quantification; the scale for cell wall fluorescence intensity is in the vertical direction, taller peaks represent greater fluorescence intensity. (**a**,**d**), (**b**,**e**), (**c**,**f**): data for CC-1690, CC-5325, and *cw15*, respectively. (**g**) Boxplot for cell wall fluorescence quantification. Total fluorescence intensity of each cell wall was normalized to the cell perimeter length. About 100 cells were quantified for each strain for fluorescence intensity. Statistical analyses were performed using a two-tailed *t*-test assuming unequal variance by comparing with CC-1690 (*, *p* < 0.05; ***, *p* < 0.001) or between CC-5325 and *cw15* (###, *p* < 0.001). (**g**) The percentages of cell wall fluorescence intensity in CC-5325 and *cw15* are relative to CC-1690.

**Figure 9 plants-11-00585-f009:**
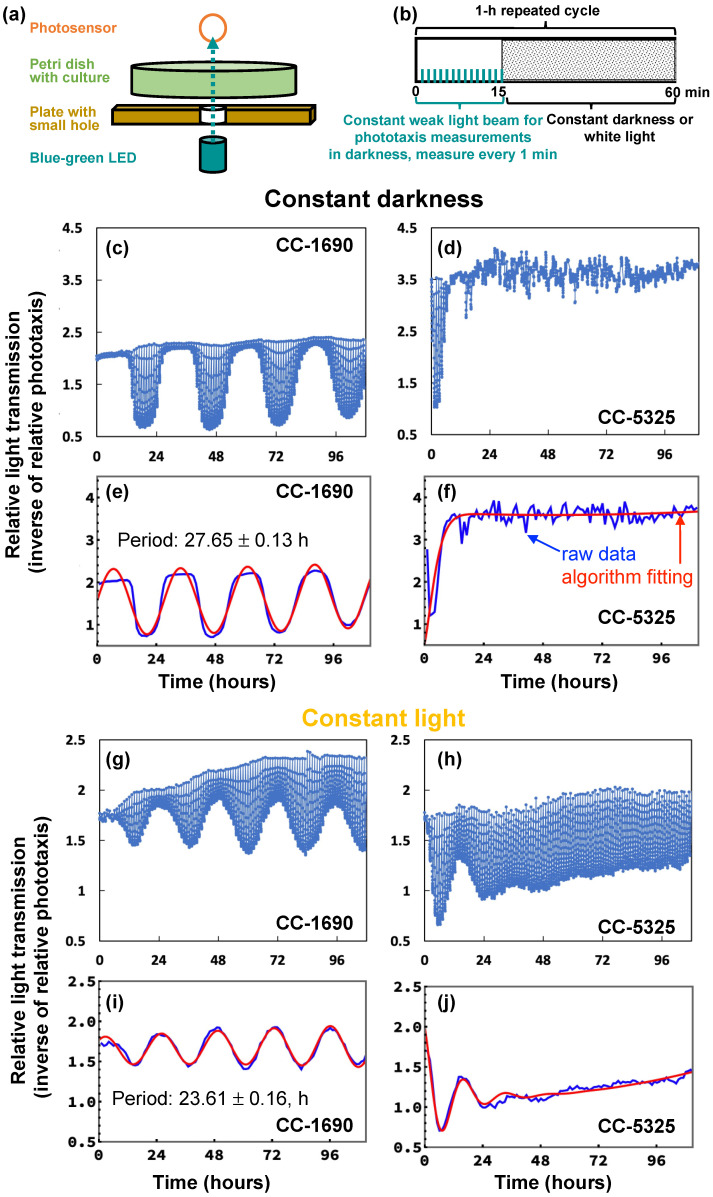
**In constant darkness or constant light, CC-1690 maintained a sustained circadian rhythm of phototaxis but CC-5325 could not.** (**a**,**b**) Schematic drawing of how the phototaxis machine collected data. Repeated every hour (h), the consecutive blue-green light-emitting diode (LED) with a maximum wavelength of 507 nm shined a narrow, weak light beam through the culture for 15 min (min) and the light transmission was recorded every min by the photosensor. In the presence of the weak light beam, cells accumulated in the beam, reducing light transmission. The more phototactically active a culture is, the more cells accumulate in the beam and reduce the amount of light transmission. For the rest of the 45 min of a 1-h-cycle, cultures are exposed to either darkness, giving rise to “constant darkness” (**c**–**f**), or white background light, giving rise to “constant light” (**g**–**j**). Graphs on the left depict data for CC-1690 (**c**,**e**,**g**,**i**) and those on the right for CC-5325 (**d**,**f**,**h**,**j**). (**c**,**d**,**g**,**h**) Example graphs of the raw data collected by the phototaxis machine. Each vertical line in the graph shows the light transmission measurements the machine collected at 1-min intervals during a 15-min period with the weak light beam every 1-h cycle. If there was no or little phototaxis at a certain period of the 15-min weak light beam, data points for that period had no or little reduction of light transmission and appeared as very short vertical lines. (**e**,**f**,**i**,**j**) The raw data from the above graphs (blue line) were fed into an analysis algorithm leading to the best-fit sine wave (red line) the algorithm determined for the raw data. The analysis algorithm only used a single data point of a 15-min weak light beam period, which is the transmission measured after 11-min into the weak light beam (when good rhythms of phototaxis were clearly shown).

**Figure 10 plants-11-00585-f010:**
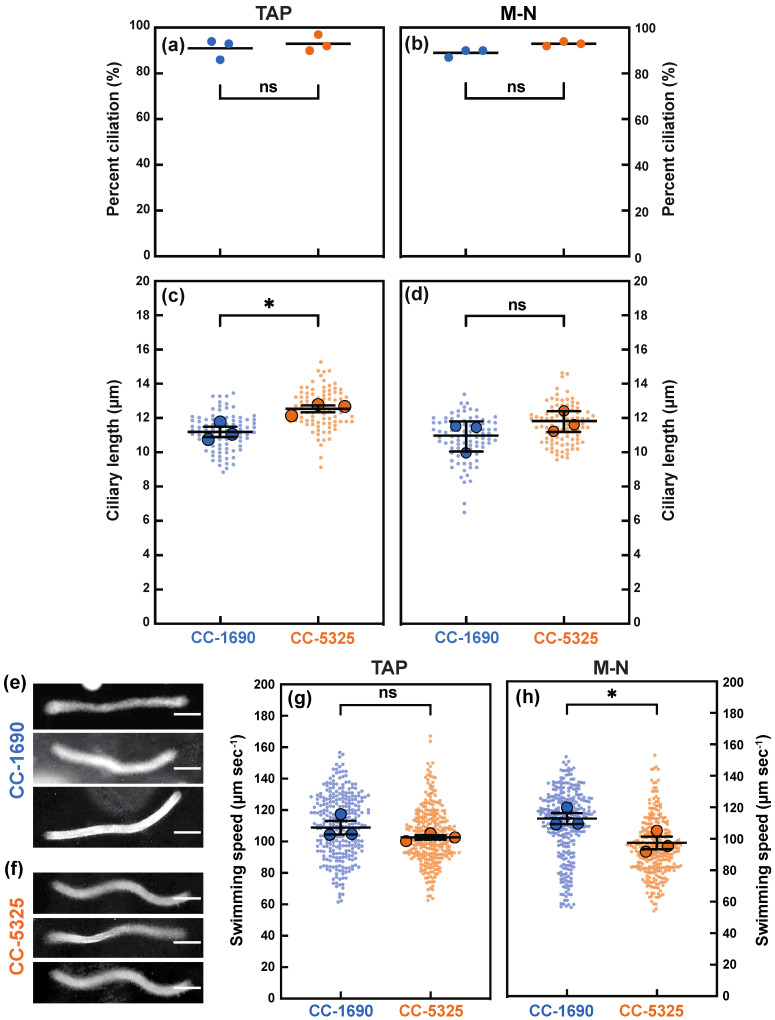
**In comparison to CC-1690, CC-5325 had longer cilia in tris-acetate-phosphate (TAP) medium but slower swimming speed in M-N medium (minimal medium without nitrogen and acetate).** CC-1690 and CC-5325 cells were grown in TAP (**a**,**c**,**g**) or in M-N (**b**,**d**,**h**) with constant agitation and under constant light of 100 μmol photons m^−2^ s^−1^. (**a**,**b**) The percentage of cells with cilia was counted. The center (horizontal lines) represents the mean of 3 biological replicates where N = 100 in each replicate. The three dots represent the mean of each biological replicate. Significance was determined by Chi-square analysis, *p* = 0.4521 for (**a**), *p* = 0.1159 for (**b**); ns, not significant. (**c**,**d**) Ciliary length was measured using ImageJ. Superplots were created by measuring the mean (the larger dots) of 3 biological replicates. Data points from individual cells are shown as small dots. The center (longer horizontal bars) represents the mean of the 3 biological replicates (N = 30 in each replicate), and the shorter horizontal bars represent standard error. Significance was determined by using a two-tailed *t*-test assuming unequal variance, (**c**) * means significance, *p* = 0.0265 and (**d**) *p* = 0.2877, ns, not significant. (**e**,**f**) Representative images of the traces created by swimming cells in TAP. CC-1690 and CC-5325 cells were allowed to swim on a microscope slide during darkfield microscopy. The camera exposure was set to 1 s (sec). The resulting images contain traces shown in the figure that represent the path the cells travelled during the 1-s exposure. The length of these traces was measured in ImageJ to estimate the swimming speed. The scale bars represent 25 µm. (**g**,**h**) The swimming speed of CC-1690 and CC-5325 cells grown in TAP (**g**) or M-N (**h**). Superplots were created by measuring the mean (the larger dots) of 3 biological replicates where N = 100 in each replicate. The center (longer horizontal bars) represents the mean of 3 biological replicates, and the shorter horizontal bars represent standard error. Significance was determined by using a two-tailed *t*-test assuming unequal variance. (**g**) *p* = 0.2422, ns, not significant. (**h**) * means significance, *p* = 0.0438.

**Figure 11 plants-11-00585-f011:**
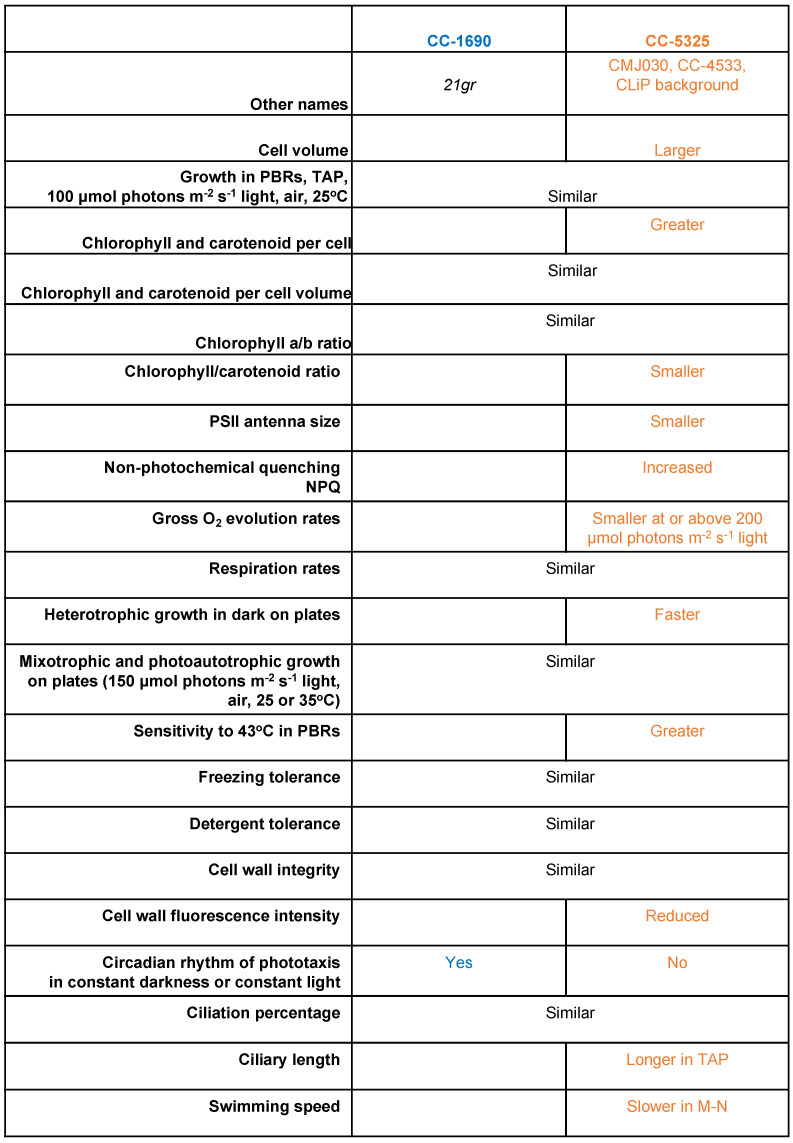
**Summary of phenotypic differences between CC-1690 and CC-5325.** TAP, Tris-acetate-phosphate medium. M-N, minimal medium without nitrogen and acetate. The phenotypic descriptions for CC-5325 are in comparison to CC-1690.

## Data Availability

The data presented in this study are available in this article.

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
