# Peer review of "Comparative Phenotyping of Two Commonly Used Chlamydomonas reinhardtii Background Strains: CC-1690 (21gr) and CC-5325 (The CLiP Mutant Library Background)"

_plants, 2022, doi:10.3390/plants11050585_

Round 1
Reviewer 1 Report
The work is rather descriptive than explorative, nevertheless, in my opinion the data presented in the manuscript can be very useful for other research groups working with Chlamydomonas because of the detailed and in-depth analysis of the strains. Experiments are clearly described, results are presented in a clear and informative way. The text is well prepared. I have no critical comments, actually.
Author Response
We thank the reviewer for the encouraging comment.
Reviewer 2 Report
The importance of Chlamydomonas as a model organism is increasing, and the CLiP mutant is one of the most important resources. The slight differences of strain CC-5325 from other laboratory strains have become an issue as the importance of the CLiP mutant increases to evaluate the mutant phenotypes. This paper is an important one that addresses this issue shared by the Chlamydomonas research community. Although the paper is rather descriptive, this is probably inevitable due to the nature of the research. The fact that the various parameters are listed numerically is worthy of this paper. However, I would like to request some corrections or additions. The major and minor points are listed below. The major point 1 is not mandatory but would add a little more value to the paper.
Major points
1.
Can you add phenotypes related to the length of cilia/flagella (nowadays, “cilia” is preferred in the Chlamydomonas community) and the number of cilia? Because photosynthesis and cilia are the two major research areas in which Chlamydomonas is used, I wonder if it would be possible to measure the percentage of cells with cilia, the length of cilia, and if possible, swimming speed, along with the cell morphology discussed in Fig. 1. This paper was submitted to Plants, and I understand that it is intended for photosynthetic researchers. Still, considering that Chlamydomonas is used as a model organism in various research fields, I think it is worth describing the above parameters of CC-5325.
2.
I could not understand which kind of experiment was done in Fig. 9. Therefore, please add a schematic diagram of the experimental system as Fig. 9A. Also, please write light conditions (wavelength, intensity) used to induce phototaxis in the figure legend.
Minor points
1.
In the Abstract, most of the parts that should be commas are semicolons, which may be an error in PDF conversion. Please check and correct them.
2.
In the second paragraph of the Introduction, at the part in which major databases for Chlamydomonas are introduced, it would be better to introduce the flagellar proteome database and the basalbody (centriole) proteome paper as well.
http://chlamyfp.org/
https://doi.org/10.1016/j.cub.2005.05.024.
3.
In the third paragraph of the Introduction, my understanding is that other strains, such as the CC-124/CC-125 pair, are used as background strains as much as, or rather more than, CC-1690. Please briefly mention why CC-1690 was used in this study.
Reviewer 3 Report
The authors describe a variety of comparative phenotypic studies to describe differences in two commonly used Chlamydomonas lab strains. The studies are thorough and look good. However, the authors overlook a major aspect of the logical comparison. They acknowledge that both strains have genome sequences available, yet they do not draw on this information in their conclusions. They find differences in phenotypes such as cell wall integrity, heterotrophic growth rates, and photosynthetic efficiency (antennae size), but fail to link these to genetic differences. The authors should add a section linking these phenotypes to genetic differences in the strains. They should describe what genes are likely to be causative for the differences they observed. This shouldn't be difficult, yet would significantly add to the work.
